# BMP9-ID1 Signaling Activates HIF-1α and VEGFA Expression to Promote Tumor Angiogenesis in Hepatocellular Carcinoma

**DOI:** 10.3390/ijms23031475

**Published:** 2022-01-27

**Authors:** Han Chen, Kouki Nio, Hong Tang, Taro Yamashita, Hikari Okada, Yingyi Li, Phuong Thi Bich Doan, Ru Li, Junyan Lv, Yoshio Sakai, Tatsuya Yamashita, Eishiro Mizukoshi, Masao Honda, Shuichi Kaneko

**Affiliations:** 1Center of Infectious Diseases, West China Hospital of Sichuan University, Chengdu 610041, China; chenhan19890801@wchscu.cn; 2Division of Infectious Diseases, State Key Laboratory of Biotherapy and Center of Infectious Diseases, West China Hospital of Sichuan University, Chengdu 610041, China; 3Department of Gastroenterology, Kanazawa University Hospital, Kanazawa 9208641, Japan; taroy@m-kanazawa.jp (T.Y.); okada0922@staff.kanazawa-u.ac.jp (H.O.); liyingyi@staff.kanazawa-u.ac.jp (Y.L.); doanphuong93@stu.kanazawa-u.ac.jp (P.T.B.D.); cnty100@stu.kanazawa-u.ac.jp (R.L.); lvjunyan@stu.kanazawa-u.ac.jp (J.L.); yoshios@m-kanazawa.jp (Y.S.); ytatsuya@m-kanazawa.jp (T.Y.); eishirom@m-kanazawa.jp (E.M.); mhonda@m-kanazawa.jp (M.H.); skaneko@m-kanazawa.jp (S.K.); 4Department of General Medicine, Kanazawa University Hospital, Kanazawa 9208641, Japan

**Keywords:** BMP9-ID1 signaling, angiogenesis, HIF-1α, VEGFA, hepatocellular carcinoma, BMP receptor inhibitor

## Abstract

Since hepatocellular carcinoma (HCC) is a typical hypervascular malignant tumor with poor prognosis, targeting angiogenesis is an important therapeutic strategy for advanced HCC. Involvement of bone morphologic protein 9 (BMP9), a transforming growth factor-beta superfamily member, has recently been reported in the development of liver diseases and angiogenesis. Here, we aimed to elucidate the role of BMP9 signaling in promoting HCC angiogenesis and to assess the antiangiogenic effect of BMP receptor inhibitors in HCC. By analyzing HCC tissue gene expression profiles, we found that BMP9 expression was significantly correlated with angiogenesis-associated genes, including HIF-1α and VEGFR2. In vitro, BMP9 induced HCC cell HIF-1α/VEGFA expression and VEGFA secretion. Silencing of the inhibitor of DNA-binding protein 1 (ID1), a transcription factor targeted by BMP9 signaling, suppressed BMP9-induced HIF-1α/VEGFA expression and VEGFA secretion, resulting in decreased human umbilical vein endothelial cell (HUVEC) lumen formation. BMP receptor inhibitors, which inhibit BMP9-ID1 signaling, suppressed BMP9-induced HIF-1α/VEGFA expression, VEGFA secretion, and HUVEC lumen formation. In vivo, the BMP receptor inhibitor LDN-212854 successfully inhibited HCC tumor growth and angiogenesis by inhibiting BMP9-ID1 signaling. In summary, BMP9-ID1 signaling promotes HCC angiogenesis by activating HIF-1α/VEGFA expression. Thus, targeting BMP9-ID1 signaling could be a pivotal therapeutic option for advanced HCC.

## 1. Introduction

Hepatocellular carcinoma (HCC) is the second-leading cause of cancer mortality worldwide [1]. The high mortality of HCC patients is partially due to the lack of effective drug therapeutic options. Since HCC is a solid tumor with a hypervascular nature, antiangiogenic therapy has been considered a vital therapeutic strategy for advanced HCC. Recently, several tyrosine kinase inhibitors (TKIs) that inhibit molecular signals to enhance angiogenesis have been developed to prolong the prognosis of patients with advanced HCC [2]. In addition, a new TKI therapeutic strategy combining immune checkpoint inhibitors with inhibitors of vascular endothelial growth factor (VEGF), which enhances HCC angiogenesis, has been developed [3,4]. These clinical advances in drug therapies suggest that angiogenic signaling in HCC is an important therapeutic target; thus, more studies to reveal the mechanism of HCC angiogenesis are needed to improve antiangiogenic therapy for HCC.

Bone morphologic proteins (BMPs) are a group of signaling molecules that belong to the transforming growth factor-beta (TGF-β) superfamily and are the major molecules involved in liver development [5]. Among BMP family members, BMP9 has recently been highlighted to be related to stem cell differentiation, angiogenesis, metabolism, fibrosis, and tumor growth in the liver [6,7,8]. In our recent study, we demonstrated that elevated BMP9 expression is associated with poorer HCC prognosis and that BMP9 promotes EpCAM-positive cancer stem cell properties in HCC by inhibiting DNA-binding protein 1 (ID1) [9]. However, the roles of BMP9-ID1 signaling in HCC angiogenesis remain unclear. Here, we demonstrate the potential role of BMP9-ID1 signaling in the activation of angiogenesis in HCC cells, and that BMP receptor inhibitors induce an antiangiogenic effect by inhibiting BMP9-ID1 signaling.

## 2. Results

### 2.1. BMP9 Expression Is Related to the Expression of Angiogenic Factors in HCC Tissues

To investigate the interaction between BMP9 and angiogenic factors in HCC, we first explored the gene expression correlation between BMP9 (*GDF2*) and the angiogenic proteins VEGFR2 (*KDR*) and HIF-1α (*HIF1A*) in The Cancer Genome Atlas Liver Hepatocellular Carcinoma (TCGA-LIHC) cohort. As expected, we observed a significant correlation between *GDF2* and *VEGFR2* (Figure 1A, R = 0.19, *p* = 0.00037) and *GDF2* and *HIF1A* (Figure 1B, R = 0.12, *p* = 0.017).

By further analyzing the correlation between BMP9 and HIF-1α in HCC tissue specimens, we found that BMP9 expression was correlated with HIF-1α expression (*p* = 0.045, Figure 1C,D, Table 1). These results showed that BMP9 expression is related to the expression of angiogenic factors in HCC tissues.

### 2.2. BMP9 Promotes HIF-1α-VEGFA Expression in HCC Cells to Enhance Vascular Endothelial Cell Activation

We next investigated the role of BMP9 in angiogenic signaling in HCC. We first examined the effect of recombinant BMP9 on the expression of HIF-1α and VEGFA in the ID1-positive HCC cell lines Huh7 and MT (*ID1* is a target gene of BMP9). Compared with the control, BMP9 upregulated the gene expression of HIF-1α and VEGFA, as well as its target gene ID1 in Huh7 and MT cells (Figure 2A). Consistent with this result, western blot analysis demonstrated that recombinant BMP9 increased the expression of pSmad1/5, ID1, HIF-1α, and VEGFA in Huh7 and MT cells in a dose-dependent manner (Figure 2B,C). Since VEGFA is a secreted protein, we also tested the VEGFA level in the cell culture supernatants of Huh7 and MT cells treated with/without BMP9 by ELISA. The results showed significantly increased VEGFA levels in the supernatants of Huh7 and MT cell lines treated with BMP9 (Figure 2D). We further tested the lumen formation of human umbilical vein endothelial cell (HUVEC) by transferring HCC cell culture medium to confirm the proangiogenic effect of elevated VEGFA (Figure 2E). In the lumen formation assay, the BMP9-treated HCC cell culture medium supernatants induced better lumen integrity and longer lumen length (Figure 2F,G). To assess whether the effects on lumen formation were mediated by VEGFA or remnant BMP9, we also treated HUVECs with BMP9 and found that BMP9 did not promote the lumen formation of HUVECs. Taken together, these data indicate that BMP9 promotes HCC angiogenesis through the promotion of HIF-1α/VEGFA expression and VEGFA secretion from HCC cells.

### 2.3. BMP9-ID1 Signaling Promotes VEGFA Secretion from HCC Cells by Regulating HIF-1α/VEGFA Signaling

We then investigated the mechanism by which BMP9 induces HIF-1α and VEGFA expression. Since VEGFA expression was found to be abundant in ID1-positive HCC cells (Appendix A), we hypothesized that ID1 regulates BMP9-induced HIF-1α and VEGFA expression. As expected, in the immunohistochemistry (IHC) staining analysis of HCC tissues, we found that HIF-1α expression was related to ID1 expression (Figure 3A,B, Table 2). In ID1 knockdown of Huh7 and MT cells, while VEGFA gene expression, but not HIF-1α gene expression, was decreased (Figure 4A), HIF-1α and VEGFA protein expression was decreased (Figure 4B). Consistent with a previous report, these data indicate that ID1 regulates HIF-1α/VEGFA expression by a mechanism different from HIF-1α gene regulation. We also found that knockdown of ID1 inhibited the secretion of VEGFA from HCC cells and the lumen formation of HUVECs (Figure 4C–E). We additionally confirmed that knockdown of ID1 reduced BMP9-induced HIF-1α/VEGFA expression and VEGFA secretion (Figure 4F–H). In contrast with ID1 knockdown, ID1 overexpression increased HIF-1α/VEGFA protein expression, VEGFA secretion, and the lumen formation of HUVECs (Figure 5A–E). Taken together, these findings indicate that BMP9-ID1 promotes HUVEC activation by regulating HIF-1α/VEGFA signaling.

### 2.4. BMP Receptor Inhibitors Suppress BMP9-ID1-Induced VEGFA Secretion

The above data indicated that BMP9-ID1 signaling could be a crucial therapeutic target for suppressing HCC angiogenesis. We thus examined the effect of the BMP receptor inhibitors K02288 and LDN-212854. Compared with the control, BMP receptor inhibitors greatly suppressed HIF-1α and VEGFA expression (Figure 6A–C) in Huh7 and MT cells. We additionally confirmed that cell culture supernatants taken from HCC cells treated with the BMP receptor inhibitor LDN-212854 inhibited the lumen formation of HUVECs (Figure 6D,E). We further evaluated the effect of BMP receptor inhibitors in the presence of BMP9. As expected, K02288 and LDN-212854 significantly suppressed the BMP9-induced upregulation of HIF-1α/VEGFA expression and VEGFA secretion (Figure 6F,G). Taken together, these findings suggest that BMP receptor inhibitors are potentially promising therapeutic agents inhibiting HCC angiogenesis.

### 2.5. BMP9 Receptor Inhibitors Suppress HCC Tumor Progression through the Repression of Angiogenic Factors In Vivo

According to all the in vitro findings, LDN-212854 was the most potent inhibitor to of HCC angiogenesis, which it suppressed by inhibiting BMP9-ID1 signaling. We thus assessed the antitumor/antiangiogenic effect of LDN-212854 on HCC xenografts. Compared with PBS, LDN-212854 significantly suppressed the growth in of tumors derived from Huh7 cells (*p* = 0.0101) and MT cells (*p* = 0.0053) in an in vivo mouse xenograft model (Figure 7A,B). In addition, LDN-212854 inhibited ID1, HIF-1α, VEGFA, and VEGFR2 expression (Figure 7C–E). These data demonstrated that the BMP receptor inhibitor LDN-212854 inhibits HCC tumor growth and angiogenic signaling by suppressing BMP9-ID1 signaling, suggesting that targeting BMP9-ID1 signaling could be a promising therapeutic option for HCC patients.

## 3. Discussion

Solid tumors are not isolated or homogeneous cell masses but are complex and composed of tumor cells, extracellular matrix, infiltrating lymphocytes, secreted factors, and blood vessels. Together, these factors constitute the tumor microenvironment (TME) [10,11]. HCC is a typical malignant tumor with a hypervascular nature, and angiogenesis is considered a key player in the TME in HCC development. Among angiogenesis stimulators, VEGFA is the center, and VEGFA is a direct target of HIF-1α. Targeting VEGFA and its upstream protein HIF-1α is thus a new strategy for antiangiogenic therapy [12,13].

BMP9, also known as GDF-2, is a member of the BMP subgroup of TGF-β superfamily proteins that signal through heterodimeric complexes composed of type I and type II BMP receptors. Recently, we reported that BMP9 is a crucial factor inducing the malignant nature of HCC and found that BMP9-ID1 signaling promotes cancer stem cell properties in EpCAM-positive HCC cells by activating Wnt/β-catenin signaling [9]. The role of BMP9 in angiogenesis is still controversial [14]; thus, we aimed to investigate the functional roles of BMP9 signaling in the activation of angiogenic signaling in HCC cells in the current study.

We first demonstrated the correlation between *GDF2* and the angiogenic genes *HIF1A*/*KDR* by analyzing the TCGA-LIHC dataset. We additionally demonstrated that HIF-1α expression is positively correlated with BMP9 expression in IHC analysis of HCC tissue specimens. These findings implied that BMP9 is involved in HCC angiogenesis. Consistent with this idea, we confirmed that BMP9 induces ID1 and HIF-1α/VEGFA expression and VEGFA secretion from HCC cells by administering recombinant BMP9 protein, which resulted in the activation of HUVEC lumen formation. Since BMP9 did not promote the lumen formation of HUVECs, this activation was thought to be induced by VEGFA secretion from HCC cells. 

We further investigated the mechanisms by which BMP9 induces HIF-1α/VEGFA expression. Interestingly, VEGFA expression was abundant in ID1-expressing HCC cells, which suggested that ID1 may be involved in the regulation of HIF-1α/VEGFA signaling. As expected, ID1 silencing/overexpression experiments revealed that ID1 regulates the expression of HIF-1α and VEGFA. It was reported that in breast cancer cell lines, ID1 induces HIF-1α/VEGFA protein expression but not HIF-1α mRNA expression; in addition, ID1 enhances the stability of the HIF-1α protein [15]. Similarly, in our present study, ID1 induced HIF-1α/VEGFA protein expression but not mRNA expression, suggesting that ID1 enhances the stability of the HIF-1α protein in HCC cells.

In our recent study, we indicated that BMP receptor inhibitors are therapeutic agents that can potentially inhibit the cancer stem cell-related processes in HCC, including tumor progression, recurrence, and metastasis, by inhibiting BMP9-ID1 signaling [9]. Thus, we further investigated the antiangiogenic effects of BMP receptor inhibitors in HCC. As expected, BMP receptor inhibitors suppressed HIF-1α/VEGFA expression and VEGFA secretion, resulting in the impairment of HUVEC lumen formation. Furthermore, we also found the involvement of another BMP9 signaling pathway in the regulation of HIF-1α/VEGFA expression via BMP receptor inhibitor treatment experiments. In contrast to ID1 silencing, BMP receptor inhibitor treatment decreased HIF-1α gene expression. This result suggested that BMP9 enhances HIF-1α/VEGFA expression by signaling upstream of ID1. It has been reported that BMP9 induces HIF-1α via the Smad transcription factors Smad1/5/8, which directly affect the HIF-1α promoter [16]. The finding that ID1-dependent/ID1-independent pathways are involved in the regulation of HIF-1α/VEGFA expression by BMP9 is very interesting. In addition, this notion suggests that BMP receptor inhibitors are more potent inhibitors of HIF-1α/VEGFA expression because they block both pathways. In line with this reasoning, we further confirmed that BMP receptor inhibitors successfully suppressed tumor growth, as well as ID1, HIF-1α, VEGFA, and VEGFR2 expression in HCC xenograft tumors.

Our study showed the crucial mechanism by which BMP9 affects HCC angiogenesis: induction of HIF-1α/VEGFA. Furthermore, we demonstrated the utility of BMP receptor inhibitors for not only inhibiting cancer stemness but also inhibiting angiogenesis in HCC. Since BMP9, which promotes cancer stemness and angiogenesis in HCC, plays an important role in tumor progression, therapy targeting BMP9 signaling could be a novel therapeutic option for advanced HCC. Hence, further research on the practical application of BMP receptor inhibitors is desirable.

## 4. Conclusions

Our study revealed that BMP9 signaling plays a crucial role in HCC angiogenesis through the activation of HIF-1α/VEGFA signaling. BMP receptor inhibitors are thus new potential therapeutic options targeting angiogenesis for patients with advanced HCC.

## 5. Materials and Methods

### 5.1. Clinical Samples

A total of 53 HCC tissue specimens were obtained after informed consent was obtained from patients who underwent liver resection at Kanazawa University Hospital, from 2011 to 2014. All subjects gave their informed consent for inclusion before they participated in the study in which the tissue would be used for future study (2016-093, 25 October 2011). The study subjects were opted out of participation in the study (2017-323, 18 April 2018) by posting the study on the hospital website (https://web.hosp.kanazawa-u.ac.jp/research/exam/) (accessed on 23 December 2021). The study was conducted in accordance with the Declaration of Helsinki, and the protocols were approved by the Ethics Committee of the Graduate School of Medical Sciences, Kanazawa University (2016-093, 2017-323).

### 5.2. Cell Lines and Reagents

The HCC cell lines Huh7, HepG2, HLE, HLF, and SK-Hep-1 and HUVEC were obtained from the Japanese Collection of Research Bioresources Cell Bank (Osaka, Japan), American Type Culture Collection (Manassas, VA, USA) or Procell Life Science & Technology (Wuhan, China). The HCC cell line MT was established from resected HCC specimens as described previously [17]. HCC cells were maintained in Dulbecco’s modified Eagle medium (DMEM; Gibco, Grand Island, NY, USA) supplemented with foetal bovine serum (FBS; Gibco) at 37 °C. HUVECs were maintained in an endothelial cell growth medium-2 (EGM-2 BulletKit; Lonza, Basel, Switzerland). The recombinant BMP9 protein was purchased from R&D Systems (Minneapolis, MN, USA) and Abcam (Cambridge, UK). The BMP receptor inhibitors K02288 and LDN-212854 were purchased from Selleck Chemicals (Houston, TX, USA) and dissolved in dimethyl sulfoxide (Sigma Aldrich, St. Louis, MO, USA).

### 5.3. RNA Interference and Plasmid Transfection

The Silencer^TM^ Select ID1-specific siRNA s7104 and negative control siRNA were purchased from Invitrogen (Carlsbad, CA, USA). Cells were cultured in culture medium without FBS for 24 h prior to siRNA transfection. The siRNA constructs were transfected using Lipofectamine RNAiMAX (Invitrogen) in accordance with the manufacturer’s protocol. At 4–6 h post-transfection, the cells were washed with PBS to completely remove the siRNA constructs in the medium and were then cultured in DMEM with 10% FBS.

PCMV6-AC-GFP-ID1 (RG202061) was purchased from Origene Technologies, Inc. (Rockville, MD, USA), and the pcDNA3.1 (V790-20) plasmid, which was used as an empty vector control, was purchased from Invitrogen. The DNA constructs were transfected using Lipofectamine 2000 (Invitrogen) in accordance with the manufacturer’s protocol. At 4–6 h post-transfection, the cells were washed with PBS to completely remove the DNA constructs in the medium and were then cultured in DMEM with 10% FBS.

### 5.4. Real-Time Quantitative PCR

Total RNA was extracted using a High Pure RNA Isolation Kit (Roche Diagnostics K.K., Tokyo, Japan) according to the manufacturer’s instructions. HIF-1α (Hs00153153_m1), VEGFA (Hs00900055_m1) and ID1 (Hs03676575_s1) quantitative PCR probes were purchased from Applied Biosystems (Foster City, CA, USA). The expression of selected genes was determined in triplicate using the 7900 Sequence Detection System (Applied Biosystems). Each sample was normalized relative to the expression of reference genes (β-actin or 18S rRNA).

### 5.5. Western Blotting

Cell lysates were extracted using RIPA buffer, and the following primary antibodies were used for Western blotting: anti-ID1 monoclonal, sc-133104, (Cell Signaling Technology, Danvers, MA, USA); anti-ID1 monoclonal, ab230679 (Abcam); anti-VEGFA polyclonal, ab46154 (Abcam); anti-VEGFA monoclonal, ab214424 (Abcam); anti-HIF-1α monoclonal, ab51608 (Abcam); anti-VEGF receptor 2 monoclonal #2479 (Cell Signaling Technology); anti-phospho-Smad1/5 (Ser463/465) monoclonal #9576 (Cell Signaling Technology); anti-Smad5 monoclonal #12534 (Cell Signaling Technology); and anti-β-actin monoclonal, #4970 (Cell Signaling Technology). Immune complexes were visualized using enhanced chemiluminescence detection reagents (Amersham Biosciences Corp., Piscataway, NJ, USA) according to the manufacturer’s instructions.

### 5.6. IHC Staining

The following primary antibodies were used for IHC staining: anti-BMP9 polyclonal, ab35088 (Abcam); anti-BMP9 polyclonal, PA5-11931 (Invitrogen); anti-ID1 sc-133104, (Cell Signaling Technology); anti-HIF-1α monoclonal, ab51608 (Abcam); and anti-VEGF receptor 2 monoclonal #2479 (Cell Signaling Technology). Envision+ kits (Dako, Carpinteria, CA, USA) were used in accordance with the manufacturer’s instructions. IHC staining images were obtained using an inverted microscope (BZ-X800, KEYENCE, OSAKA, Japan) and a Zeiss microscope (Zeiss, Oberkochen, Germany).

### 5.7. Lumen Formation Assay

The lumen formation assay was assessed in 96-well plates coated with growth factor-reduced Matrigel (BD Biosciences, Franklin Lakes, NJ, USA) as described previously [18]. HUVECs (3 × 10^4^ cells/well) were seeded on top of Matrigel (50 µL/well)-coated wells in EGM-2 medium. When capillary-like structures (lumens) formed, the upper medium was replaced by different kinds of Huh7 or MT cell media. The lumen structures on the Matrigel were recorded after incubation for 24 h at 37 °C using a phase-contrast microscope (BZ-X800, KEYENCE). The total lumen area was quantified as the total lumen length from the image analysis of the microscopic field using the BZ-X800 Analyzer Program (KEYENCE). The data are presented as the means ± SD of the total lumen length per field.

### 5.8. Animal Studies

NOD.CB17-*Prkdc^scid^*/J (NOD/SCID) mice were purchased from Charles River Laboratories, Inc. (Wilmington, MA, USA). Mice were housed under specific pathogen-free conditions with a 12 h light/dark cycle and provided ad libitum access to tap water and food. Huh7 or MT cells (1 × 10^6^ cells) were resuspended in 200 µL of a 1:1 DMEM:Matrigel (BD Biosciences) mixture and subcutaneously injected into 4- to 6-week-old NOD/SCID mice. Once tumors had reached a measurable size, mice were randomly divided into two groups (each *n* = 3) and intraperitoneally injected with PBS or 6 mg/kg LDN-212854 twice daily for 10–14 days. The size of subcutaneous tumors was recorded at study termination. The experimental protocol was approved by the Kanazawa University Animal Care and Use Committee (AP-183972; 26 March 2021) and conformed to the Guide for the Care and Use of Laboratory Animals prepared by the National Academy of Sciences.

### 5.9. Analysis of the TCGA-LIHC Dataset

The correlation of *GDF2* (BMP9) with *KDR* (VEGFR2) and *GDF2* (BMP9) with *HIF1A* (HIF-1α) was investigated using the TCGA-LIHC dataset obtained from cBioportal (https://www.cbioportal.org/ (accessed on 23 December 2021)).

### 5.10. Statistical Analysis

Overall survival analysis, Student’s *t* test, one-way ANOVA, the chi-square test, Fisher’s exact test, and the log-rank test were performed using GraphPad Prism 8 (GraphPad Software, San Diego, CA, USA). A *p* value of less than 0.05 was considered significant.

## Figures and Tables

**Figure 1 ijms-23-01475-f001:**
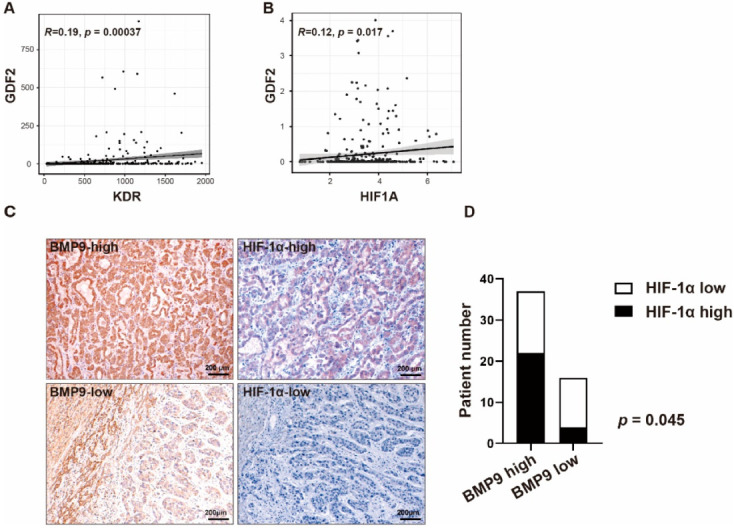
Bone morphologic protein 9 (BMP9) expression is correlated with the expression of angiogenesis-associated factors. (**A**) Correlation between BMP9 (*GDF2*) and VEGFR2 (*KDR*) gene expression. (**B**) Correlation between BMP9 (*GDF2*) and HIF-1α (*HIF1A*) gene expression. (**C**) Representative immunohistochemistry (IHC) staining images of BMP9/HIF-1α high (upper panel) and BMP9/HIF-1α low (lower panel) surgically resected hepatocellular carcinoma (HCC) tissue specimens. (**D**) Correlation of BMP9 and HIF-1α in HCC patients. The *p* value was calculated by the chi-squared test.

**Figure 2 ijms-23-01475-f002:**
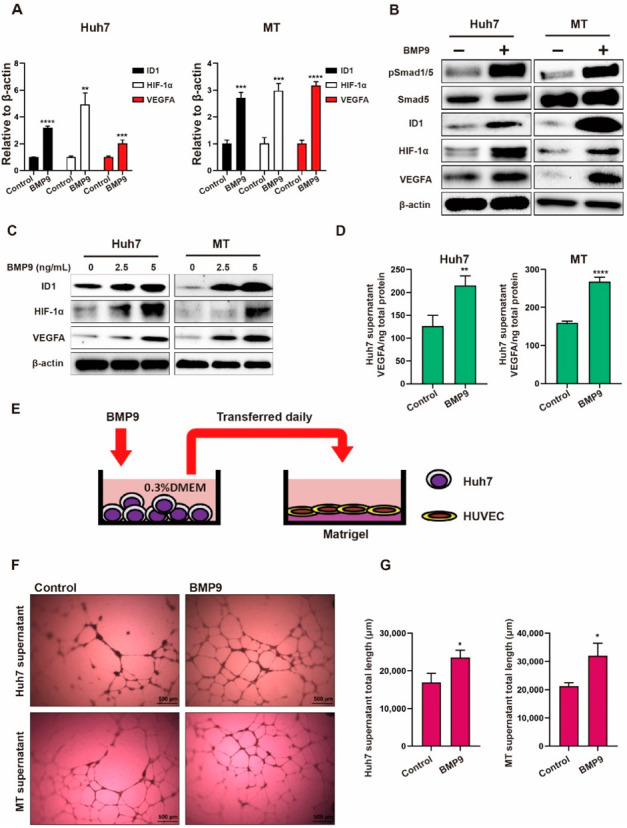
BMP9 activates HIF-1α/VEGFA signaling in HCC cells. (**A**) Relative gene expression levels of inhibiting DNA-binding protein 1 (ID1), HIF-1α and VEGFA in Huh7 and MT cells. Cells were treated with dimethyl sulfoxide or BMP9 (5 ng/mL) for 48 h. (**B**) Western blot analysis of pSmad1/5, Smad5, ID1, HIF-1α and VEGFA expression in Huh7 and MT cells. Cells were treated with dimethyl sulfoxide or BMP9 (5 ng/mL) for 48 h. (**C**) Western blot analysis of ID1, HIF-1α and VEGFA expression in Huh7 and MT cells treated with different concentrations of BMP9 for 48 h. (**D**) ELISA analysis of VEGFA in the supernatant of Huh7 and MT cell cultures. Cells were treated with dimethyl sulfoxide or BMP9 (5 ng/mL) for 48 h. (**E**) Schematic of the lumen formation assay. (**F**) Representative pictures of the lumen formation assay of human umbilical vein endothelial cell (HUVEC)l culture supernatants were collected from Huh7 or MT cells treated with dimethyl sulfoxide or BMP9 (5 ng/mL) for 48 h. HUVECs were treated with these supernatants for 24 h. (**G**) Lumen length analysis of the lumen formation assay. The error bars represent the SD from at least three independent biological replicates. Student’s *t* test was used to calculate *p* values, represented as * *p* < 0.05; ** *p* < 0.01; *** *p* < 0.001; **** *p* < 0.0001.

**Figure 3 ijms-23-01475-f003:**
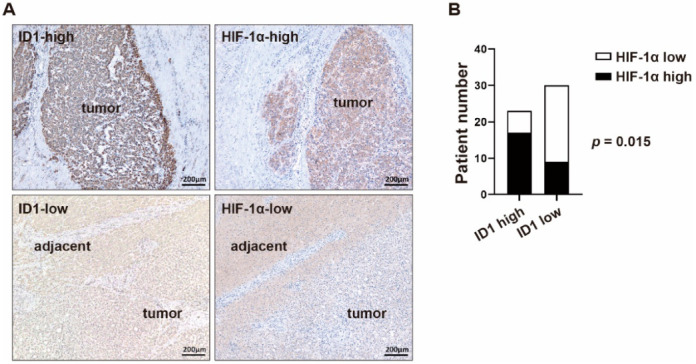
The correlation of ID1 and HIF-1α in clinical samples. (**A**) Representative IHC staining images of ID1/HIF-1α-high (upper panel) and ID1/HIF-1α-low (lower panel) surgically resected HCC tissue specimens. (**B**) Correlation of ID1 and HIF-1α in HCC patient samples. The *p* value was calculated by the chi-squared test.

**Figure 4 ijms-23-01475-f004:**
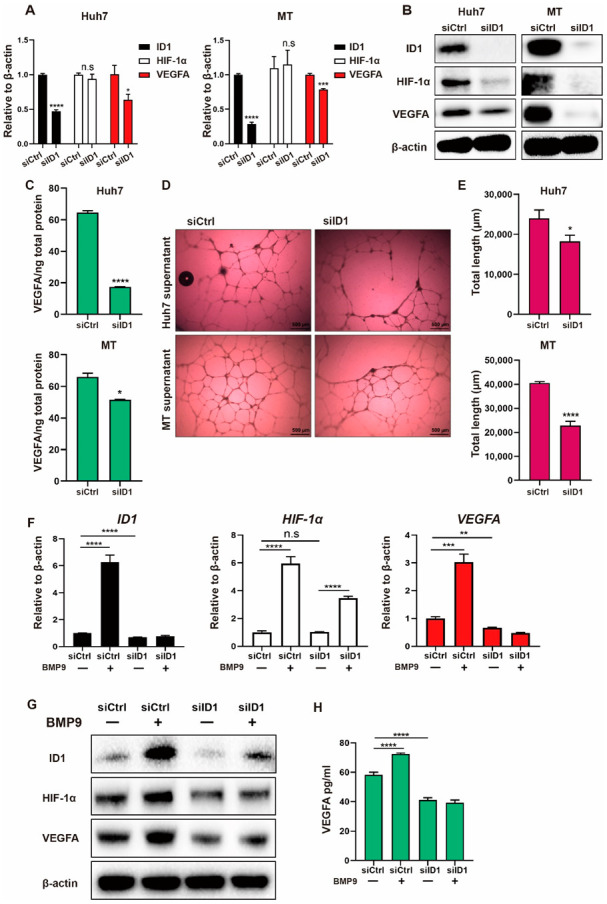
Inhibition of ID1 represses BMP9-induced HIF-1α/VEGFA signaling activity in HCC cells. (**A**) Relative gene expression levels of ID1, HIF-1α and VEGFA in Huh7 and MT cells. Cells were harvested following transfection with 20 nM control siRNA (siCtrl) or ID1 siRNA (siID1) for 48 h. (**B**) Western blot analysis of ID1, HIF-1α and VEGFA in Huh7 and MT cells. Cells were harvested after transfection with 20 nM control siRNA (siCtrl) or ID1 siRNA (siID1) for 48 h. (**C**) ELISA analysis of VEGFA in the cell culture supernatants of Huh7 and MT cells. Cell culture supernatants were harvested following transfection with 20 nM control siRNA (siCtrl) or ID1 siRNA (siID1) for 72 h. (**D**,**E**) Lumen formation assay of HUVECs treated with cell culture supernatants from Huh7 and MT cells. Cell culture supernatants were harvested following transfection with 20 nM control siRNA (siCtrl) or ID1 siRNA (siID1) for 72 h. (**F**) Relative gene expression levels of ID1, HIF-1α and VEGFA in Huh7 cells. Huh7 cells were treated with or without BMP9 (5 ng/mL) for 48 h following siRNA transfection. (**G**) Western blot analysis of ID1, HIF-1α and VEGFA in Huh7 cells. Huh7 cells were treated with or without BMP9 (5 ng/mL) for 48 h following siRNA transfection. (**H**) ELISA analysis of VEGFA in Huh7 cell culture supernatants. Huh7 cells were treated with or without BMP9 (5 ng/mL) for 48 h following siRNA transfection. The error bars represent the SD from at least three independent biological replicates. Student’s *t* test was used to calculate *p* values, represented as * *p* < 0.05; ** *p* < 0.01; *** *p* < 0.001; **** *p* < 0.0001; n.s, not significant.

**Figure 5 ijms-23-01475-f005:**
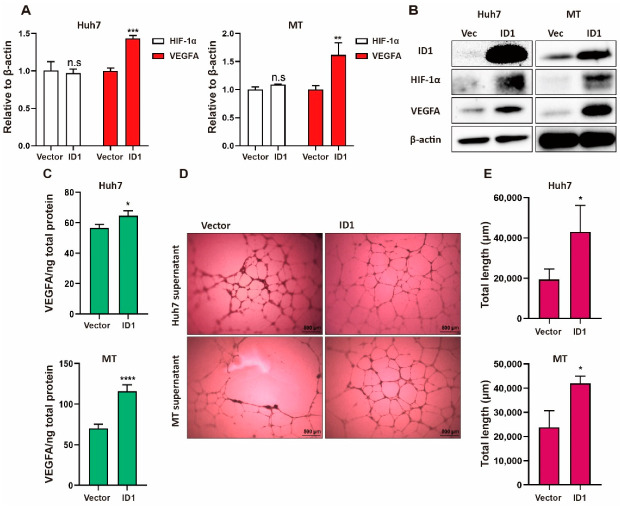
Overexpression of ID1 promotes HIF-1α/VEGFA signaling activity in HCC cells. (**A**) Relative gene expression levels of HIF-1α and VEGFA in Huh7 and MT cells. Cells were harvested following transfection with 1 μg vector or ID1 plasmid for 48 h. (**B**) Western blot analysis of ID1, HIF-1α and VEGFA in Huh7 and MT cells. Cells were harvested following transfection with 1 μg vector or ID1 plasmid for 48 h. (**C**) ELISA analysis of VEGFA in Huh7 and MT cell culture supernatants. Cell culture supernatants were collected after the cells were transfected with 1 μg vector or ID1 plasmid for 48 h. (**D**,**E**) Lumen formation assay of HUVECs treated with cell culture supernatants from Huh7 and MT cells for 24 h. Cell culture supernatants were harvested following transfection with 1 μg vector or ID1 plasmid for 48 h. Student’s *t* test was used to calculate *p* values, represented as * *p* < 0.05; ** *p* < 0.01; *** *p* < 0.001; **** *p* < 0.0001; n.s, not significant.

**Figure 6 ijms-23-01475-f006:**
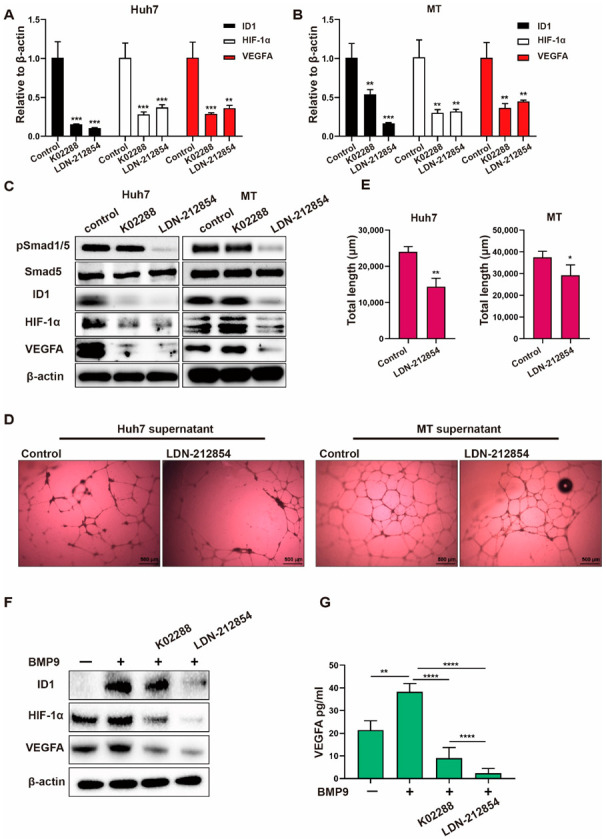
BMP receptor inhibitors repress BMP9-induced HIF-1α/VEGFA signaling activity in HCC cells. (**A**,**B**) Relative gene expression levels of ID1, HIF-1α and VEGFA in Huh7 and MT cells. Cells were harvested after being treated with 1 μM BMP receptor inhibitors for 48 h. (**C**) Western blot analysis of pSmad1/5, Smad5, ID1, HIF-1α and VEGFA in Huh7 and MT cells. Cells were harvested after being treated with 1 μM BMP receptor inhibitors for 48 h. (**D**,**E**) Lumen formation assay of HUVECs treated with supernatants from Huh7 and MT cells for 24 h. Cell supernatants were harvested following treatment with 1 μM BMP receptor inhibitors or 10 nM ramucirumab. (**F**) Western blot analysis of ID1, HIF-1α and VEGFA in Huh7 cells treated with 1 μM BMP receptor inhibitors in the presence of BMP9 (5 ng/mL) for 48 h. (**G**) ELISA analysis of VEGFA in Huh7 supernatants. Supernatants were collected after cells were treated with 1 μM BMP receptor inhibitors in the presence of BMP9 (5 ng/mL) for 48 h. The error bars represent the SD from at least three independent biological replicates. Student’s *t* test was used to calculate *p* values, represented as * *p* < 0.05; ** *p* < 0.01; *** *p* < 0.001; **** *p* < 0.0001.

**Figure 7 ijms-23-01475-f007:**
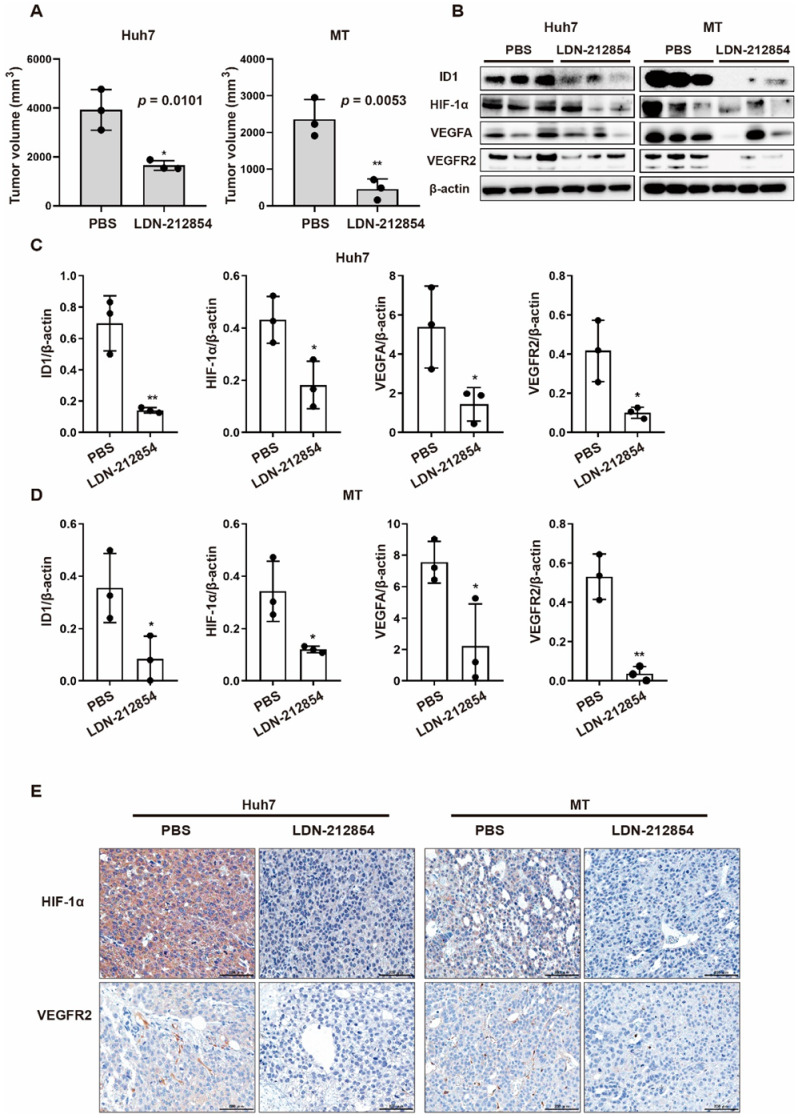
The BMP receptor inhibitor LDN-212854 suppresses the angiogenesis of HCC xenografts through repression of ID1 in vivo. (**A**) Effect of LDN-212854 on the growth of Huh7 and MT cells. Mice were sacrificed after being injected with PBS or 6 mg/kg LDN-212854 twice daily for 10 days (Huh7 xenograft tumors) or 14 days (MT xenograft tumors). (**B**) Xenograft tumors of Huh7 and MT cells treated with PBS (*n* = 3) or LDN-212854 (*n* = 3). (**C,D**) Western blot analysis of ID1, HIF-1α, VEGFA, and VEGFR2 expression in xenograft tumors derived from Huh7 and MT cells. β-Actin was used as the reference for quantifying protein expression. (**E**) Representative IHC staining images of HIF-1α and VEGFR2 expression in Huh7 and MT cell xenografts. The error bars represent the SD from at least three independent biological replicates. Student’s t test was used to calculate *p* values, represented as * *p* < 0.05; ** *p* < 0.01.

**Table 1 ijms-23-01475-t001:** BMP9 and HIF-1α expression in HCC patient samples.

	BMP9 High (*n* = 37)	BMP9 Low (*n* = 16)
HIF-1α high	22	4
HIF-1α low	15	12

*p* = 0.045.

**Table 2 ijms-23-01475-t002:** ID1 and HIF-1α expression in HCC patient samples.

	ID1 High (*n* = 23)	ID1 Low (*n* = 30)
HIF-1α high	17	9
HIF-1α low	6	21

*p* = 0.015.

## Data Availability

The TCGA-LIHC dataset is publicly available. The data that support the findings of this study are available in the relevant figures/tables of this article.

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
