# Peer review of "BMP9-ID1 Signaling Activates HIF-1α and VEGFA Expression to Promote Tumor Angiogenesis in Hepatocellular Carcinoma"

_ijms, 2022, doi:10.3390/ijms23031475_

Round 1

Reviewer 1 Report

Chen et. al study the role of BMP9-ID1 signaling in the activation of the angiogenesis in hepatocellular carcinoma and the anti-angiogenic effect of BMP receptor inhibitors. The methods employed are adequate in order to clarify the under-study mechanisms. Moreover, the results obtained are of great interest. This manuscript can be accepted for publication after minor revisions. In particular, authors must address the following issues:

  1. All Figures must be re-formatted and be provided in better resolution. There is a major problem of similarity in Figs and authors must use the same fonts/font size in all axes and inset legends. Furthermore, in Fig. 4A,C,E,F – Fig.5A, C – Fig.6A,B,E the legends at x-axes are not clear.

         2. The manuscript suffers from major grammatical and syntax English    errors that requires to be addressed. Furthermore, the employed phraseology is inconsistent with the formality required for a scientific journal. For example, phrases like “Given the data“, “Considering our data“, “Given all our findings“, “Given clinical data“, etc. must be removed.

Reviewer 2 Report

General Comments: 

Dr. Han Chen and colleagues propose a research article concerning the role of BMP9-ID1 in tumor angiogenesis in hepatocellular carcinoma through the expression of HIF-1α/VEGFA proposing BMP9-ID1 signaling to be a pivotal therapeutic option for advanced HCC.

In general, the article is well conducted and the findings are well presented. I did not identified any major flaws in the methodology used,  no major flaws in the data presented, no misleading or false conclusions. I recommend therefore the article to be accepted in your journal for publication.

Minor comments:

I would recommend the authors to provide details of all the sequences of primers used for qRT-PCR as well as the sequence of RNA interference used in the materials and methods.
